# Progranulin Protects against Hyperglycemia-Induced Neuronal Dysfunction through GSK3β Signaling

**DOI:** 10.3390/cells12131803

**Published:** 2023-07-07

**Authors:** Cass Dedert, Lyuba Salih, Fenglian Xu

**Affiliations:** 1Department of Biology, College of Arts and Sciences, Saint Louis University, Saint Louis, MO 63103, USA; cass.dedert@slu.edu (C.D.); lyuba.salih@slu.edu (L.S.); 2Institute for Translational Neuroscience, Saint Louis University, Saint Louis, MO 63103, USA; 3Department of Pharmacology and Physiology, School of Medicine, Saint Louis University, Saint Louis, MO 63103, USA; 4Department of Biomedical Engineering, School of Science and Engineering, Saint Louis University, Saint Louis, MO 63103, USA

**Keywords:** diabetes, hyperglycemia, progranulin, GSK3β, autophagy, neurons

## Abstract

Type II diabetes affects over 530 million individuals worldwide and contributes to a host of neurological pathologies. Uncontrolled high blood glucose (hyperglycemia) is a major factor in diabetic pathology, and glucose regulation is a common goal for maintenance in patients. We have found that the neuronal growth factor progranulin protects against hyperglycemic stress in neurons, and although its mechanism of action is uncertain, our findings identified Glycogen Synthase Kinase 3β (GSK3β) as being potentially involved in its effects. In this study, we treated mouse primary cortical neurons exposed to high-glucose conditions with progranulin and a selective pharmacological inhibitor of GSK3β before assessing neuronal health and function. Whole-cell and mitochondrial viability were both improved by progranulin under high-glucose stress in a GSK3β—dependent manner. This extended to autophagy flux, indicated by the expressions of autophagosome marker Light Chain 3B (LC3B) and lysosome marker Lysosome-Associated Membrane Protein 2A (LAMP2A), which were affected by progranulin and showed heterogeneous changes from GSK3β inhibition. Lastly, GSK3β inhibition attenuated downstream calcium signaling and neuronal firing effects due to acute progranulin treatment. These data indicate that GSK3β plays an important role in progranulin’s neuroprotective effects under hyperglycemic stress and serves as a jumping-off point to explore progranulin’s protective capabilities in other neurodegenerative models.

## 1. Introduction

Uncontrolled hyperglycemia, or elevated blood glucose, is a major characteristic of type II diabetes (T2D), a metabolic disease that affects over 530 million individuals worldwide [1]. Left unchecked, hyperglycemia contributes to a host of diabetic pathologies, including metabolic, nephropathic, and neuropathic diseases [2,3,4]. There is an increased prevalence of neurodegenerative diseases like Alzheimer’s and Parkinson’s among diabetic patients [5,6], and animal models of T2D show impaired neuronal health and cognitive function [7,8,9]. Despite advances in understanding the development of T2D and its associated sequelae, a definitive treatment for hyperglycemia-induced neurodegeneration remains elusive.

Common to T2D and several neurodegenerative diseases is the dysregulation of autophagy, a cellular self-degradation pathway involved in turnover of aggregated and damaged proteins [10,11]. In neurons, autophagy is a major pathway for protein clearance due to their post-mitotic nature [12]. The downregulation of autophagy leads to protein buildup and cell death, and the subsequent rescue of autophagy flux reduces aggregate buildup, improving cell health [13]. T2D impairs autophagy through the upregulation of growth-related signaling kinases such as mammalian Target Of Rapamycin Complex 1 (mTORC1) [14] and the downregulation of autophagy-inducing kinases such as AMP-activated protein Kinase (AMPK) [15] and GSK3β [16]. Upregulating autophagy function under hyperglycemic stress presents a promising treatment modality for T2D and has been utilized in diabetic medications like metformin [17], but a more comprehensive treatment method has not been elaborated.

Progranulin (PGRN) is a 68 kDa growth factor with neuroprotective properties [18] that is highly expressed in neurons and microglia, although it is present in most cell types [19]. Patients with a hereditary form of the neurodegenerative frontotemporal lobar dementia (FTLD) have mutations in *GRN* [20,21,22], and progranulin knockout in cell and animal models leads to neurodegeneration and cognitive impairment [23,24]. On the other hand, an overexpression of PGRN in models of Alzheimer’s disease leads to improved health outcomes, suggesting a neuroprotective effect [25]. PGRN is trafficked to the lysosome via Sortilin, where it is cleaved into granulin subunits that serve protease functions [18]. It also has protective functions independent of Sortilin [26], indicating that the full scope of its activity lies beyond the lysosome. However, the exact mechanism of its action has yet to be conclusively determined.

Since a common underlying cause of pathology in neurodegenerative diseases is impaired autophagy flux [11,12], it is possible that PGRN protects against neuronal damage through the preservation of this metabolic pathway, among others. However, it is not fully determined whether PGRN also attenuates the downregulation of autophagy under diabetic conditions. While previous research in our lab has provided evidence indicating that this is the case and that GSK3β phosphorylation decreased following PGRN treatment [27], the exact mechanism of action has been unexplored. We sought to continue that line of inquiry in this study, exploring the hypothesis that GSK3β is a mechanistic cause of PGRN’s neuroprotection. GSK3β’s activity is implicated in autophagy signaling and is normally downregulated by hyperglycemic stress [16,28,29], making this an appealing protein for investigation. There is prior research suggesting an interplay between PGRN and GSK3β, although the activation/inhibition of the latter is inconclusive and may be brain-region-dependent [30,31,32]. However, it has not been explored whether PGRN is implicated in neuroprotection against high glucose stress, and whether GSK3β is involved in that process.

In this study, we provide evidence that GSK3β appears to play a role in PGRN’s neuroprotection against high glucose through the preservation of cell development, regulation of autophagy flux, and moderation of neuronal firing activity. These findings provide a mechanistic framework for PGRN’s protective action in the face of hyperglycemic stress and serve a foundation for future studies investigating PGRN’s neuroprotection in other neurodegenerative models.

## 2. Materials and Methods

### 2.1. Animals and Cell Culture

Experiments were performed on cortical neurons dissected from P0-P3 C57BL/6 mouse pups, which were removed under sterile conditions according to the standard protocol approved by the Institutional Animal Care and Use Committee (IACUC), Saint Louis University, St. Louis, MO, guidelines. Cortices were dissected and incubated in an enzymatic solution containing 40 units of papain (Worthington Biochemical, Cat# LS003126), 2 mM CaCl_2_ (Sigma, Cat# C4901-100G), 1 mM EDTA (Sigma, Cat# E9884-100G), and 1.5 mM L-cysteine (Sigma, Cat# 168149-25G) in neurobasal medium (Gibco, Cat# 21103-049). Tissues were incubated in enzymatic solution for 30 min at 37 °C, mixing every few minutes to ensure even digestion. After incubation, the enzymatic solution was inactivated using neurobasal complete medium, i.e., medium supplemented with 1X GlutaMAX (Gibco, Cat# 35050-061), 1% pen/strep (Gibco, Cat# 15140122), 2% B-27 supplement (Gibco, Cat# 17504-044), and 4% fetal bovine serum (Avantor, Cat# 97068-086). Tissues were then triturated through fire-polished glass pipettes, then plated on dishes coated with 2 µg/mL laminin (Sigma, Cat# 11243217001) and 100 µg/mL poly-D lysine (Sigma, Cat# P6407-5MG). Cells were cultured in neurobasal complete medium, and one half of the medium was changed every three to four days.

### 2.2. Hyperglycemia, Progranulin, and Inhibitor Treatment

Treatment with high glucose was achieved in primary cortical cells by supplementing neurobasal complete medium with glucose (Sigma, Cat# G6152) to reach a final concentration of 100 mM, with a control medium containing 25 mM glucose. This is consistent with ours and previous studies in which control primary cells are cultured at 25 mM glucose [27,33]. Recombinant mouse progranulin (PGRN, R&D Systems, Cat# 2557-PG050) dissolved in PBS was added to achieve a final concentration of 2.94 nM, consistent with our previous studies in vitro [27]. The pharmacological inhibition of GSK3β was achieved using 10 µM of the GSK3 inhibitor SB216763 (Selleckchem, Cat# S1075) dissolved in DMSO. Where appropriate, treatment with 0.1% DMSO was used as a vehicle control. Cells were treated for 72 h before performing the experiments listed below, except for calcium imaging experiments, in which cells were pre-treated for 1 h before recording.

### 2.3. Determination of Cell Viability and Neurite Outgrowth

The status of cell health and morphology was viewed using a phase-contrast microscope (IX73, Olympus) and images were acquired with a Retiga R1 camera (QImaging) and the ImageJ plugin Micro-Manager (UCSF). Cell viability was determined with a fluorescence-based reporter dye kit (LIVE-DEAD^TM^ Cell Imaging Kit, ThermoFisher, Cat# R37601). After treatment, cells were washed thrice with PBS, then incubated in HBSS (Gibco, Cat# 14025-076) containing 1 µM Calcein AM and 2 µM ethidium homodimer at 37 °C for 45 min. Images were taken using an inverted microscope (IX73, Olympus) equipped with fluorescence light filters (Lambda XL, Sutter Instrument) and a Retiga R1 camera. Cell viability was determined by manually counting the number and calculating the ratio of ethidium-homodimer-stained cells to total cells (i.e., those stained with either Calcein AM or ethidium homodimer) in each image. The neurite number and status (primary, secondary, or tertiary) were recorded using the ImageJ plugin NeuronJ (NIH).

### 2.4. Determination of Mitochondrial Viability

The status of mitochondrial health was viewed by incubating treated cells with the cell-permeable dye tetramethylrhodamine (TMRM; Invitrogen, Cat# T668), which is taken up in proportion to mitochondrial membrane potential. After treatment, cells were washed with PBS, and then incubated in HBSS containing 100 nM TMRM dissolved in DMSO at 37 °C for 45 min. After incubation, cells were immediately imaged under a fluorescence microscope equipped with fluorescence light filters (IX73, Olympus) with the fluorescent light delivered by Lambda XL (Sutter Instrument). The mitochondrial membrane potential was determined by measuring the mean fluorescence intensity of TMRM in neuronal regions of interest (ROIs), and mean fluorescence intensity was normalized so that the control group was equal to 1.0.

### 2.5. Mitochondrial Complex Enzyme Activity Assays

Mitochondrial activity was tested in whole-cell lysates of protein harvested after treatment, with protocols used in the prior literature [27,34,35]. All measurements were tied to a change in absorbance over time, measured in a 96-well microplate format using a plate reader (Synergy H1, BioTek) and standardized to sample protein concentration. The activities of ubiquinone oxidoreductase (UO), succinate dehydrogenase (SDH), cytochrome C oxidoreductase (CCO), and cytochrome C oxidase (COX) were tested to represent mitochondrial complexes I, II, III, and IV, respectively. All reagents listed were obtained from Sigma unless otherwise noted.

For UO activity, samples were added to a reagent containing 25 mM potassium phosphate, pH 7.2 (Cat# P5655), 5 mM MgCl_2_ (Cat# M4880), 1 mM KCN (Cat# 60178), 0.13 mM NADH (Cat# N8129), 65 μM decylubiquinone (Cat# D7911-10MG), 2.5 mg/mL BSA (Cat# A9418), and 2 μg/mL antimycin A (Cat# A8764). The reagent was heated to 30°C for 10 min before adding 2 μg/mL rotenone (Cat# R8875), followed by adding samples. The activity was tied to the reduction of NADH, measured as a decrease in absorbance at 340 nm over a 20 min period.

For SDH activity, samples were added to a reagent containing 10 mM KCl (Cat# P5405), 5 mM MgCl2, 50 mM sodium succinate (Cat# S2378), 40 mM NaN_3_ (Cat# S2002), 300 mM mannitol (Cat# M4125), and 20 mM potassium phosphate (pH 7.2). The activity was tied to the reduction of the electron acceptor DPIP (Fisher Chemical, Cat# S286-5) (50 μM), which manifests as a decrease in absorbance at 600 nm over a 30 min period.

For CCO activity, samples were added to a reagent containing 50 mM potassium phosphate, pH 7.2, 1 mM n-dodecyl-β-D-maltoside (Cat# D4641), 2 mM KCN, and 20 mM NADH. The reagent was heated to 30°C for 10 min before adding 100 μM decylubiquinone, then 40 μM oxidized cytochrome C (Cat# C2506), followed by adding samples. The activity was tied to reduction of cytochrome C, measured as an increase in absorbance at 550 nm over a 30 min period.

For COX activity, samples were added to a reagent containing 20 mM potassium phosphate, pH 7.2, and 0.45 mM n-dodecyl-β-D-maltoside. The reagent was heated to 30°C for 10 min before adding 15 μM reduced cytochrome C, followed by adding samples. The activity was tied to oxidation of cytochrome C, measured as a decrease in absorbance at 550 nm over a 30 min period.

### 2.6. Immunofluorescence

Primary cortical cells were fixed with 4% paraformaldehyde (ThermoFisher, Cat# J19943-K2) for 20 min, permeabilized with 0.3% Triton X-100 (VWR, Cat# 0694-1L) for 5 min, and blocked in PBS containing 5% goat serum (Gibco, Cat# 16210-064) for 1 h. Primary antibodies for LC3B (CST, Cat# 43566S), LAMP2A (Abcam, Cat# ab18528), microtubule-associated protein 2 (MAP2) (Invitrogen, Cat# 13-1500), and GSK3β (CST, Cat# 9315S) were used at a 1:200 dilution in 5% goat serum. The following secondary antibodies were used at a 1:500 dilution in 5% goat serum: goat anti-rabbit conjugated with Alexa Fluor 568 (Invitrogen, Cat# A11036), and goat anti-mouse conjugated with Alexa Fluor 488 (Invitrogen, Cat# A11029). The slides were stained using Fluoroshield mounting media containing 1 µg/mL DAPI (Sigma, Cat# F6507). The images were taken using a TCS SP8 confocal microscope (Leica).

Fluorescence intensity analysis was performed by selecting regions of interest (ROIs) of neuronal somata, identified through staining with MAP2. The mean fluorescence intensity of each ROI was measured using ImageJ software, and values were normalized to set the mean control equal to 1. All images were taken under the same acquisition parameters to prevent differences due to user error.

### 2.7. Immunoblotting

Samples were treated with Laemmli sample buffer (Bio-Rad, Cat# 1610747), treated with 350 mM dithiothreitol (DTT; Bio-Rad, Cat# 1610611), and heated to 95 °C for 5 min before running. Samples were run on 4–12% bis-tris gels (NuPage, Cat# NP0323BOX) using a Novex Mini-Cell device (Invitrogen, Cat# EI0001). The gels were then transferred to 0.45 µm nitrocellulose membranes (Bio-Rad, Cat# 1620115) in a Mini Protean Tetra System (Bio-Rad, Cat# 1658004).

Membranes were blocked with 5% milk in TBST, then blotted using primary antibodies for p-Akt ser473 (CST, Cat# 9271S), Akt (CST, Cat# 9272S), p-S6K thr389 (CST, Cat# 9234S), S6K (CST, Cat# 2708S), p-Unc-Like Kinase 1 (ULK1) ser757 (CST, Cat# 6888S), ULK1 (CST, Cat# 8054S), Transcription factor EB (TFEB; Proteintech, Cat# 13372-1-AP), and Glyceraldehyde 3-phosphate dehydrogenase (GAPDH; CST, Cat# 2118S). Antibodies for Akt, S6K, and ULK1 were generously provided by the lab of Dr. Jonathan Fisher. All primary antibodies were used at a 1:1000 dilution. After an overnight incubation at 4 °C, membranes were incubated in a secondary antibody at a 1:5000 dilution for one hour (goat anti-rabbit conjugated with horseradish peroxidase, Invitrogen, Cat# 31460), then washed and incubated in Pierce enhanced chemiluminescent substrate (ThermoFisher, Cat# 32106). Western blot data were collected using an imager (iBright FL1000, ThermoFisher) and densitometric analysis was performed using ImageJ (NIH).

### 2.8. Calcium Imaging

Calcium influx was measured using a Fura 2-AM ester (Invitrogen, Cat# F1221), which binds selectively to calcium. The cells were pre-treated with 10 µM SB216763 or 0.1% DMSO (vehicle control) for 1 h prior to experimentation. The cells were washed twice with HBSS, then incubated in HBSS containing 5 µM Fura 2-AM at 37 °C for 30 min. After incubation, the cells were washed four times with HBSS before imaging. Fura-2 was excited at its excitation maxima (340 nm and 380 nm) sequentially via a Lambda XL instrument equipped with a high-speed wavelength switcher (Sutter Instrument), then imaged with a microscope equipped with a Retiga R1 camera (QImaging). MetaFluor imaging software (version 7.8.2.0, Molecular Devices, LLC) was used to acquire fluorescence intensity, with a higher ratio of 340 nm to 380 nm indicating greater calcium influx. Baseline readings were collected for 2–3 min before acute treatment with 2.94 nM PGRN; readings were collected for 2–3 min after. At the end of the experiments, cells were treated with 10 mM KCl as a positive control to verify neuronal status. The background was subtracted from fluorescence intensity values for cells using an empty ROI as a negative control. The degree of calcium influx due to acute treatment was measured as the maximum increase from the mean baseline 340/380 nm ratio (peak Δ 340/380 nm ratio). Only cells that responded to KCl with a sharp upward spike in calcium influx (a trait specific to neurons) were included for analysis.

### 2.9. Multielectrode Array (MEA)

Cells were cultured in 60-electrode multielectrode array (MEA, Multichannel Systems, Cat# 60MEA200/30iR-Ti) and pre-treated with 10 µM SB216763 or 0.1% DMSO (vehicle control) for 1 h prior to experimentation. Neuronal activities were tested in an MEA2100-Lite workstation (Multichannel Systems), with the temperature kept at 37 °C using a TC 01 temperature controller. The frequency of spontaneous electrical spikes was recorded for five minutes before treatment (i.e., baseline) and for five minutes after treatment with 2.94 nM PGRN using Multi Channel Experimenter software. Data were analyzed using a Multi Channel Analyzer and Multi Channel Data Manager software (Multichannel Systems), and spike frequencies were compared pre- and post-treatment within individual neurochips.

### 2.10. Statistical Analysis

Data were analyzed using GraphPad Prism 9.0 software, with the threshold for significance set to *p* < 0.05. The values provided are of the mean S.E.M. An unpaired Student’s *t*-test was used to compare significance between two groups (except for MEA recordings, which were assessed with a paired Student’s *t*-test). To compare more than two groups, one-way ANOVA was used. Post hoc testing was performed through Fisher’s Least Significant Difference (LSD). In cases where statistical deviations were significant (assessed through Brown–Forsythe and Bartlett’s tests), Welch’s ANOVA was used to determine significance between more than two groups, and post hoc testing was performed using Student’s *t*-test with Welch’s correction.

## 3. Results

### 3.1. Neuronal Cell Viability under High-Glucose Stress Is Preserved by PGRN in a GSK3β-Dependent Manner

GSK3β is downregulated by mTORC1 and Wnt signaling, both of which are upregulated in diabetic conditions [36,37]. Because of our prior findings suggesting a decrease in inhibitory GSK3β phosphorylation, we explored whether the pharmacological inhibition of GSK3β attenuates PGRN’s beneficial effects, using the ATP-competitive, GSK3-specific inhibitor SB216763 [38]. Our results show that treatment with 10 µM SB216763 (a concentration used in several other studies exploring GSK3β signaling [39,40]) had deleterious effects on neuronal viability and development in cells treated with high glucose and PGRN, in addition to a buildup of proteinaceous debris around the soma (see HG + PGRN + SB, Figure 1A). These experiments revealed a significant decrease in cell viability due to high glucose and GSK3β inhibition (F = 6.221, *p* = 0.001). The percent viability decreased due to high glucose from 86.49 ± 1.93% to 78.17 ± 2.01% (*p* = 0.004), and while PGRN co-treatment significantly increased the viability from 78.17 ± 2.01% to 88.97 ± 1.52% (*p* = 0.000), the inhibition of GSK3β lowered the viability rate from 88.97 ± 1.52% to 81.89 ± 2.12% (*p* = 0.007) (Figure 1B). Viability changes were not present in cells treated with SB216763 alone (Appendix A), suggesting that GSK3β inhibition under high glucose stress led to this phenotype, even when cells were co-treated with PGRN.

Neurite quantity, a metric for neuronal development, improved with PGRN under high-glucose conditions, with SB216763 treatment attenuating this effect (W = 3.552, *p* = 0.002) (Figure 1C). High glucose reduced the neurite number from 8.154 ± 0.723 neurites to 7.348 ± 0.415 neurites, although this was not significant compared to control levels (*p* = 0.339). PGRN treatment increased the neurite number under high-glucose conditions from 7.348 ± 0.415 neurites to 9.913 ± 0.820 neurites (*p* = 0.009 between HG and HG + PGRN). SB216763 co-treatment with HG + PGRN reduced the total number of neurites from 9.913 ± 0.820 neurites to 8.160 ± 0.423 neurites. While this was only a trend towards a statistically significant decrease from HG + PGRN (*p* = 0.066), it was also no longer significantly increased from HG + SB216763 (*p* = 0.059), suggesting that PGRN has less of an effect in promoting neurite outgrowth when GSK3β is inhibited. Interestingly, we found that the attenuation of neurite number due to SB216763 under HG + PGRN conditions occurred in non-primary (i.e., secondary and tertiary) but not primary neurites. These data indicate that GSK3β activity is involved in PGRN’s neuroprotective and neurite development roles in early-developing cortical neurons.

### 3.2. Neuronal Mitochondrial Health and Activity Is Preserved by PGRN in a GSK3β-Dependent Manner under High-Glucose Stress

The maintenance of mitochondrial dynamics is especially crucial in neurons due to their high metabolic requirements and post-mitotic nature, making them vulnerable to excessive ROS generation downstream of mitochondrial activity [41]. We studied mitochondrial health through measuring their membrane potential (ΔΨ_m_) in cortical cells using the fluorescence-based dye TMRM. The degree of red fluorescence intensity (which is taken up in proportion to ΔΨ_m_) increased with high glucose with and without PGRN, with SB216763 negating this increase (W = 4.383, *p* = 0.006) (Figure 2A). Short-term (72 h) treatment with high glucose significantly increased TMRM fluorescence from 1.000 ± 0.018 AU to 1.189 ± 0.066 AU; *p* = 0.0), and this elevation was maintained with PGRN co-treatment (from 1.000 ± 0.018 AU to 1.107 ± 0.045 AU (*p* = 0.029). However, SB216763 treatment alongside HG + PGRN lowered ΔΨ_m_ to a control level, a significant decrease from HG + PGRN conditions (from 1.107 ± 0.045 AU to 0.989 ± 0.022 AU; *p* = 0.020).

Diabetic pathology leads to a shift in metabolic function, leading to impaired mitochondrial function over time, a major source of neurodegeneration [42,43]. We explored whether this is the case in our neuronal hyperglycemic model and if PGRN protects against this through GSK3β by looking at the activity of rate-limiting enzymes of the mitochondrial complexes. Microplate assays of cortical cells revealed heterogenous changes in mitochondrial complex activity (Figure 2B–E). The activity of ubiquinone oxidoreductase (UO, complex I) did not change (F = 0.635, *p* = 0.601). Cytochrome C oxidoreductase (CCO, complex III) activity trended towards significance (W = 3.421, *p* = 0.069), with an apparent decrease in the HG + PGRN and HG + PGRN + SB216763 groups compared to control. Succinate dehydrogenase (SDH, complex II) activity was significantly affected (F = 3.650, *p* = 0.030), with a trend towards a significant increase with HG + PGRN compared to high glucose alone (from 36.109 ± 3.743 ΔmOD_600_ to 53.472 ± 7.164 ΔmOD_600_, *p* = 0.056), and with the level of activity significantly decreasing due to SB216763 treatment (from 53.472 ± 7.164 ΔmOD_600_ to 25.824 ± 5.767 ΔmOD_600_, *p* = 0.004). Lastly, cytochrome C oxidase (COX) activity decreased significantly (F = 4.340, *p* = 0.043), with a trend towards reduced activity when PGRN was added in high-glucose conditions (from 202.927 ± 18.619 ΔmOD_600_ to 110.434 ± 27.687 ΔmOD_600_, *p* = 0.082).

In summary, the combination of high glucose and PGRN may affect mitochondrial health and function at multiple steps of the electron transport chain, and overall membrane potential may be affected by GSK3β inhibition.

### 3.3. PGRN-Mediated Autophagy Flux under High Glucose Involves GSK3β Activity

We and other studies have presented data indicating that PGRN affects autophagy and protein turnover, but the exact mechanism of its effect on autophagy flux has been unexplored as-of-yet. We treated primary cortical neurons with high glucose, PGRN, and SB216763 for 72 h before performing immunofluorescence analysis with LC3B and LAMP2A, markers for the autophagosome and lysosome, respectively. Under high glucose, PGRN, and SB216763 treatment, fluorescence intensity changed significantly for both LC3B (W = 19.21, *p* = 0.000) and LAMP2A (F = 11.09, *p* = 0.000).

While neurons treated with high glucose showed no change in LC3B fluorescence intensity (from 1.000 ± 0.038 to 0.902 ± 0.065 AU, *p* = 0.217), those treated with PGRN had a significantly greater LC3B fluorescence under high glucose, from 0.902 ± 0.065 to 1.740 ± 0.109 AU (*p* = 0.000). SB216763 co-treatment reduced this fluorescence significantly, from 1.740 ± 0.109 to 0.694 ± 0.083 AU (*p* = 0.000) (Figure 3A,B; see Appendix A for full-size images). While SB216763 treatment decreased LC3B fluorescence under all treatment conditions except high glucose, the degree of significance was greater in those cells treated with PGRN than those without (Appendix A). We not only saw a decrease in somatic LC3B expression due to PGRN and SB216763 co-treatment, but an apparent decrease in neuritic LC3B expression in cells treated with PGRN compared to those treated with PGRN + SB216763 (Figure 3C).

LAMP2A fluorescence intensity significantly increased with high glucose alone (from 1.000 ± 0.096 to 1.282 ± 0.068 AU, *p* = 0.009), although fluorescence decreased with PGRN under high glucose (from 1.282 ± 0.068 to 0.808 ± 0.059 AU, *p* = 0.000) to a level of fluorescence similar to control levels (*p* = 0.088) (Figure 4A,B). LAMP2A fluorescence intensity did not change with SB216763 co-treatment (from 0.808 ± 0.059 to 0.738 ± 0.062 AU, *p* = 0.526) (Figure 4). Interestingly, SB216763 treatment only significantly changed LAMP2A fluorescence under PGRN treatment alone (from 1.218 ± 0.091 to 0.455 ± 0.029 AU, *p* = 0.000) (Appendix A).

Collectively, these data indicate that PGRN mediates autophagy flux even under high-glucose stress, and that GSK3β activation differentially affects autophagosomal and lysosomal expression in this mechanism of action.

### 3.4. mTOR and Akt Pathway Phosphorylation Do Not Appear to Be Affected by PGRN under High Glucose in Cortical Cells

Diabetic pathology affects GSK3β phosphorylation and autophagy flux through the modulation of several different signaling pathways, notably mTORC1 and Akt. These affect important downstream mediators of autophagy, including TFEB, which promotes the expression of many autophagy proteins at the transcriptional level [44]. We therefore explored if our cells showed changes in mTOR activity through the phosphorylation of their downstream substrates: p-S6K ser389 and ULK1 ser757 (for mTORC1) and p-Akt ser473 (for mTORC2, also a prerequisite for Akt activation) [37]. Additionally, we looked at total levels of TFEB compared to GAPDH as a loading control. Our results surprisingly showed that treatment with high glucose for 72 h did not affect the cortical phosphorylation of Akt (F = 1.620, *p* = 0.260), ULK1 (F = 0.438, *p* = 0.728), or S6K (F = 1.202, *p* = 0.369) significantly. However, the overall TFEB levels decreased significantly with SB216763 under HG + PGRN treatment, dropping from 1.042 ± 0.371 AU to 0.129 ± 0.020 AU (*p* = 0.011) (Figure 5).

These data indicate that, at least in the relatively short-term, GSK3β activity does not appear to be affected by PGRN in the cortex via upstream Akt or mTOR activation.

### 3.5. PGRN-Mediated Changes in Calcium Influx and Neuronal Firing Are Affected by GSK3β

Calcium is an important second messenger for which the cytosolic concentration is kept low relative to the external environment and to internal vesicles, such as the ER and mitochondria [45,46,47]. This makes intracellular calcium levels a sensitive marker for signaling activity that can be modulated in real time through voltage- and ligand-gated channels located on the cell and organelle membranes [48]. Because of its neurotrophic and neuroprotective properties, we investigated if PGRN acutely affects neuronal firing and intracellular calcium, and if these effects are maintained under high-glucose stress.

We found that PGRN acutely affects intracellular calcium uptake, as cells treated with PGRN showed an acute increase in calcium intake (measured as an increase in the peak fluorescent intensity of 340/380 nm ratio, or Δ340/380 nm) following administration (W = 38.97, *p* = 0.000). This increase was blunted when cells were pre-treated with SB216763 an hour before experimentation (Figure 6A), with the peak ratio decreasing from 0.207 ± 0.046 to 0.065 ± 0.014 Δ340/380 nm (*p* = 0.007) (Figure 6D). Interestingly, neurons incubated in high glucose for 72 h beforehand exhibited a significant increase in intracellular calcium levels from acute PGRN treatment (Figure 6B), with the peak 340–380 nm ratio increasing from 0.207 ± 0.046 to 0.450 ± 0.068 340/380 nm (*p* = 0.005); prior inhibition with SB216763 did not significantly affect PGRN-mediated calcium influx (0.450 ± 0.068 to 0.597 ± 0.054 Δ340/380 nm, *p* = 0.096) (Figure 6D). These changes are due solely to acute PGRN treatment, as the baseline 340/380 nm ratio was unchanged prior to acute treatment (F = 1.136, *p* = 0.338) (Figure 6C).

Considering the importance of electrical signaling (i.e., action potentials) to the function of neurons and overall brain tissue, we investigated whether PGRN affects neuronal firing. Multi-electrode array (MEA), a neurochip-based method of study, was used to collect data on network activity in primary cortical cultures. We observed a decrease in neuronal spike frequency when primary cortical cultures were acutely treated with PGRN, from 1.000 ± 0.151 AU under baseline conditions to 0.439 ± 0.067 AU after treatment with PGRN (*p* = 0.000). Intriguingly, this effect was attenuated when cells were pre-treated with 10 µM SB216763 for 1 h prior to experimentation; when normalized, the spike frequency decreased from 1.000 ± 0.205 AU to 0.554 ± 0.290 AU after treatment with PGRN (*p* = 0.190) (Figure 6E). Cells incubated in high glucose showed no significant change in spike frequency due to PGRN (1.000 ± 0.343 AU to 0.880 ± 0.285 AU, *p* = 0.783), but SB216763 pre-treatment led to a significant decrease in spike events due to PGRN (from 1.000 ± 0.241 AU to 0.102 ± 0.022 AU, *p* = 0.000).

These data suggest that the ability for PGRN to acutely regulate calcium levels and spontaneous firing in neurons is reduced when GSK3β is inhibited. In high glucose conditions, however, GSK3β inhibition negates the PGRN-mediated regulation of firing frequency, but not intracellular calcium.

## 4. Discussion

High glucose affects a multitude of signaling pathways through both up- and down-regulation; when chronic, this change in activation profile contributes to dysfunctional cell health and function. One such kinase that has been shown to be downregulated under nutrient excess conditions is GSK3β due to inhibitory phosphorylation downstream of the nutrient sensor mTORC1 [49,50]. Since GSK3β is an important kinase for glycogen regulation and autophagy initiation [51], and because of our observed decrease in GSK3β phosphorylation following PGRN treatment [27], we further investigated the influence of this kinase on PGRN’s neuroprotective function. In this study, we found that PGRN’s role in regulating neuronal protection and activity involves functional GSK3β, at least in part, as pharmacological inhibition attenuated PGRN’s effects under high-glucose stress conditions.

The outgrowth of neurites (i.e., axons and dendrites) is crucial for neuronal development, and is promoted by growth factors such as PGRN. We and other groups have found that high glucose results in diminished neuronal development [27,52], and that PGRN promotes increased neurite outgrowth even under high-glucose stress. We therefore investigated whether this function of PGRN is mediated through GSK3β activation. Early developmental neurons were found to exhibit an increased number of total neurites when treated with PGRN under high-glucose conditions, with SB216763 treatment attenuating this increase (Figure 1C). Interestingly, we found that the negation in neurite number increase only applied to non-primary (i.e., secondary and tertiary) neurites, while having no significant effect on primary neurite number (Appendix A). This may be due in part to developmental time, as GSK3β inhibition leads to increased growth while activation leads to increased differentiation in neuronal populations [53,54]. However, studies on the role of GSK3β in neurodevelopment have yielded heterogenous results: GSK3β inhibition has been associated with increased axonal outgrowth in some studies (although this may be indirectly due to inhibitory phosphorylation via Akt [55,56]), while, in other cases, its inhibition negatively regulates axon outgrowth [57,58]. Notably, Gao et al. observed increased neurite outgrowth due to PGRN that was prevented by GSK3β knockdown [59]; however, they also observed increased GSK3β phosphorylation, but it is uncertain if this was at its inhibiting serine 9 site or activating tyrosine 216 site [60]. Collectively, these indicate that PGRN requires GSK3β activity for developing cortical neurons’ neurites, particularly in the development of sub-branches of neuronal processes.

Important to the development of neurodegenerative diseases is the imbalance of homeostasis between protein accumulation and degradation. Autophagy is an essential pathway for the clearance of long-lived and damaged proteins and organelles, and is especially crucial for the long-term survival of post-mitotic cells such as neurons [12]. It is well-established that the pathological etiology of several neurodegenerative diseases (Alzheimer’s, FTLD) involves an excessive buildup of protein aggregates and the downregulation of autophagy, which would normally remove these aggregates [11]. Diabetes is not only a risk factor that contributes to neurodegenerative disease, but also directly results in the accumulation of protein aggregates in the forms of IAPP (amylin) [61] and advanced glycation end-products (AGEs) [62]. Through immunofluorescence imaging, we demonstrated changes in autophagy flux through autophagosomal (LC3B) and lysosomal (LAMP2A) expression in neurons cultured under high glucose and PGRN (Figure 3 and Figure 4). Additionally, we observed differential profiles in autophagosome and lysosome accumulation, with SB216763 not changing the overall lysosome intensity while lowering autophagosome intensity. LC3B fluorescence was highly prevalent in both the soma and neurites with PGRN treatment, although this appeared to be diminished with GSK3β inhibition via SB216763 (Figure 3C). Since autophagy involves a number of steps, this suggests that GSK3β may be important in the later stages of autophagy, in which autophagosomes tagged with lipidated LC3B travel to the perinuclear region, fuse with lysosomes, and are degraded [63]. In this context, GSK3β may be involved in either fusion, degradation, or both. The fact that the LC3B fluorescence was decreased may also indicate decreased autophagy initiation due to GSK3β inhibition [16,64]. In line with this, we found that overall levels of the transcription factor TFEB, important in autophagy activation, were decreased with SB216763 co-treatment, even in the face of HG + PGRN treatment (Figure 5C). In the context of neurodegenerative disease due to impaired autophagy, this suggests that GSK3β may help to promote autophagy not only through the activation of ULK1, but also through preserving levels of transcription factors like TFEB that lead to the upregulation of autophagy-promoting genes. Further exploring the interaction between GSK3β and TFEB as another potential mechanism for autophagy regulation by this kinase is warranted.

Mitochondrial health is of particular importance in neurons because of their high metabolic requirements [43,65]. When mitochondria are damaged, their components are salvaged to maintain their health, preventing a prolonged compromise in membrane potential (ΔΨ_m_), cytochrome C leak, and apoptotic signaling [66]. We found that PGRN promotes increased ΔΨ_m_ in a GSK3β—dependent manner, possibly preventing cell death through maintaining this homeostasis (Figure 2A). GSK3β is involved in mitochondrial health and function outside of mitophagy, with inhibition increasing ROS generation and mitochondrial damage following injury [67]. While there is relatively little evidence directly linking GSK3β inhibition to mitophagy, it is of note that mitochondrial damage is a precursor to mitophagy as mitochondrial damage downstream of impaired ΔΨ_m_ enables the accumulation of Parkin on the outer mitochondrial membrane [68].

Several pathway axes that inhibit GSK3β play a role in development and homeostasis, including PTEN-Akt-GSK3β [30,69] and mTORC1-GSK3β [50]. We and other studies have shown decreased GSK3β phosphorylation following PGRN treatment in different neural cell types [27,31,32]. Interestingly, mice treated with valproic acid as a model of autism spectrum disorder have a decrease in cerebellar GSK3β phosphorylation, which was reversed by PGRN treatment in an Akt-dependent manner [30]. However, we observed no significant changes in cortical Akt and mTORC1 activation due to PGRN (Figure 5A,B). This may be due in part to the different brain regions studied as well as the relatively short-term nature of our high glucose treatment. A long-term investigation of high glucose in vitro would help to elucidate whether our observations are due to treatment duration.

The overall health of neural tissues lies not just in the absence of pathological signs, but in their ability to effectively function as electrical messengers of stimuli. To assess whether neuronal function is affected by PGRN treatment, we explored multiple modes of inquiry to ascertain the acute effects of PGRN on neuronal signaling. With acute PGRN, cytosolic calcium influx increased; as an important consequence of action potential firing, this indicates that PGRN may promote acute neuronal activity (Figure 6C). Interestingly, we found that GSK3β inhibition largely muted this calcium influx, potentially indicating a role for this kinase upstream of other signaling pathways mediated by PGRN. However, exposure to high glucose increased PGRN-mediated calcium influx, and SB216763 pre-treatment did not affect this. To add to this, acute PGRN treatment decreased neuronal firing frequency, as evidenced by MEA analysis, with GSK3β inhibition preventing this firing decrease (Figure 6E). It is possible that this seeming disparity between calcium influx and neuronal firing may be due to the specific neuronal subpopulation(s) being activated: for example, acute PGRN administration may preferentially increase the function of inhibitory neuronal firing rather than decrease the firing of excitatory neurons, resulting in lower overall network activity detected by MEA. Deeper research using specific blockers of certain neuronal subpopulations (e.g., GABAergic, glutamatergic, etc.) is necessary to elucidate if a specific type of neuron is preferentially affected by PGRN. Calcium ions also serve as a second messenger for several signaling pathways (PKC, non-canonical Wnt, etc.) [45], and it is possible that PGRN’s action may work to modulate these pathways. PKC activation is well known to increase in diabetic models, and since canonical PKC isotypes require calcium as well as PLC to activate, it is possible that increased PKC activation under these conditions could potentially make up for the inhibition of GSK3β under high glucose [70,71].

PGRN is increasingly understood as an important neuroprotective peptide, although the fullest extent of its protective properties remains enigmatic despite decades of research. The data we have presented, taken collectively, provide evidence that the serine-threonine kinase GSK3β may be involved in at least some of its protective properties in cortical neurons as a potential upstream key signaling molecule. A further exploration of PGRN-GSK3β signaling in neurodegenerative models, such as those for Alzheimer’s Disease, will be greatly valuable in fully understanding the mechanism behind PGRN’s protective role. Additionally, more research in other brain regions implicated in learning and memory such as the hippocampus [72,73], as well as investigations into neuronal subpopulations and glial cell types in these regions (such as astrocytes and microglia), is necessary to verify its mechanism of action in these cell types.

## Figures and Tables

**Figure 1 cells-12-01803-f001:**
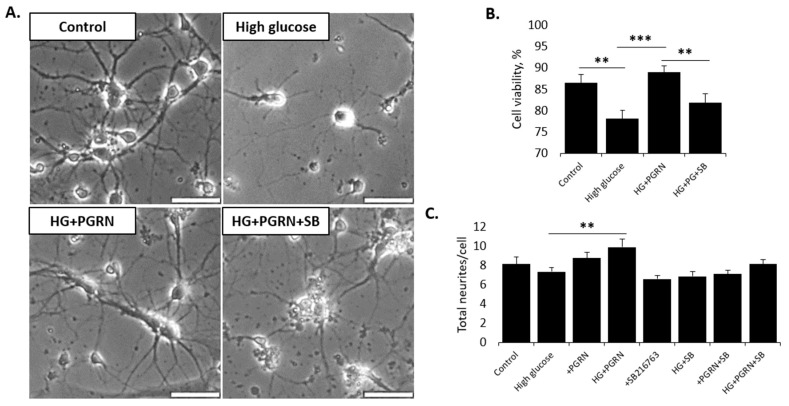
Neuronal cell viability and morphology under high-glucose stress is preserved by PGRN in a GSK3β—dependent manner. Cells were treated for 72 h with high glucose (HG, 100 mM glucose), 2.94 nM PGRN, and 10 µM SB216763 (SB) before imaging and testing. (**A**) Representative phase-contrast images of neurons after 72 h of treatment. Scale bar, 10 µm. (**B**) Cell viability, as measured by percent of cells stained with Calcein AM, decreased under high glucose, with PGRN negating this decrease. SB216763 alongside HG + PGRN decreased viability to levels seen in high glucose. N = 20–25 fields of view from two experimental replicates. (**C**) Neurite number per cell was counted at DIV4 using the ImageJ plugin NeuronJ. Total neurite number increased with PGRN treatment under high-glucose conditions, an effect that was attenuated with SB216763 co-treatment. N = 17–26 neurons from two experimental replicates. **, *p* < 0.01; ***, *p* < 0.001.

**Figure 2 cells-12-01803-f002:**
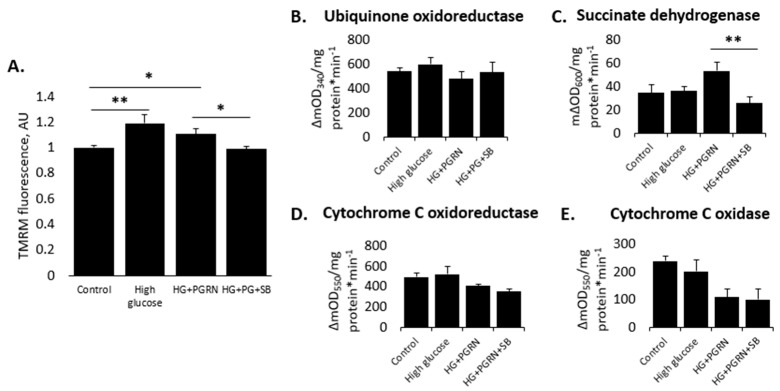
Neuronal mitochondrial health and activity are preserved by PGRN in a GSK3β—dependent manner under high-glucose stress. Cells were treated for 72 h with high glucose (HG, 100 mM glucose), 2.94 nM PGRN, and 10 µM SB216763 (SB) before harvest and/or testing. (**A**) Mitochondrial membrane potential (ΔΨ_m_), measured as TMRM fluorescence intensity, increased with high glucose and PGRN compared to control conditions. SB216763 co-treatment significantly reduced ΔΨ_m_ compared to HG + PGRN. N = 54–99 cells from two experimental replicates. (**B**–**E**) Microplate assay testing of primary cortical cells revealed heterogenous changes in UO (complex I, B), SDH (complex II, C), CCO (complex III, D), and COX (complex IV, E) activity. N = 3–6 samples from two experimental replicates. *, *p* < 0.05; **, *p* < 0.01.

**Figure 3 cells-12-01803-f003:**
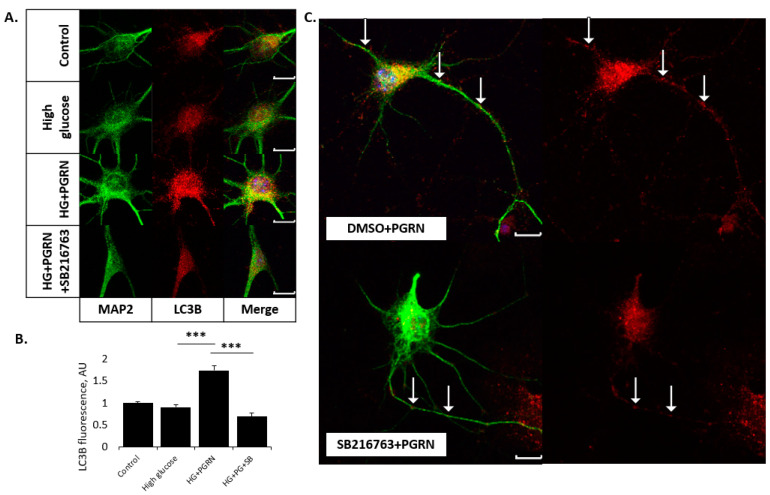
PGRN-mediated regulation of autophagosome expression in high glucose is influenced by GSK3β. Cells were treated for 72 h with high glucose (HG, 100 mM glucose), 2.94 nM PGRN, and 10 µM SB216763 (SB) before fixing and imaging. (**A**,**B**) LC3B fluorescence intensity (red) increased with PGRN under HG conditions, an effect that was attenuated with SB216763 co-treatment. Scale bar, 10 µm. N = 9–14 neurons from two experimental replicates. (**B**) Data analysis of images in A. (**C**) Representative images of neurons treated with PGRN and SB216763 + PGRN. Neurons in the latter appeared to have a lower level of LC3B fluorescence intensity (red) both in the soma and in neurites (arrows). Scale bar, 10 µm. ***, *p* < 0.001.

**Figure 4 cells-12-01803-f004:**
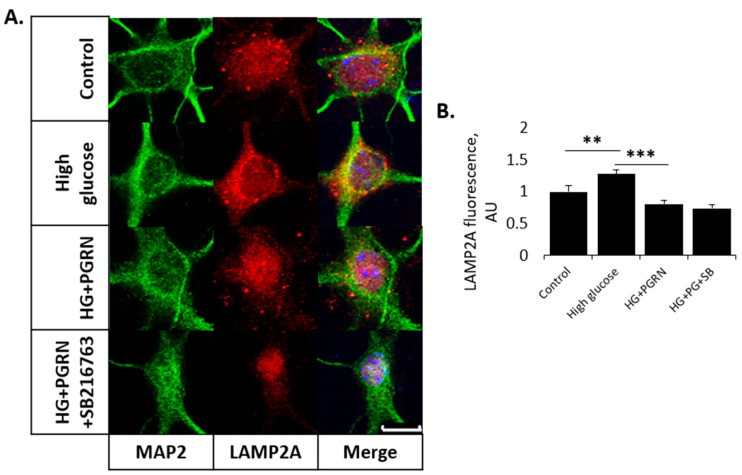
PGRN-mediated regulation of lysosome expression is not influenced by GSK3β under high glucose. Cells were treated for 72 h with high glucose (HG, 100 mM glucose), 2.94 nM PGRN, and 10 µM SB216763 (SB) before fixing and imaging. (**A**) LAMP2A fluorescence intensity (red) increased with HG, an effect that was negated with PGRN co-treatment. SB216763 did not affect LAMP2A intensity further. Scale bar, 10 µm. (**B**) Data analysis of images in A. N = 10–14 neurons from two experimental replicates. **, *p* < 0.01; ***, *p* < 0.001.

**Figure 5 cells-12-01803-f005:**
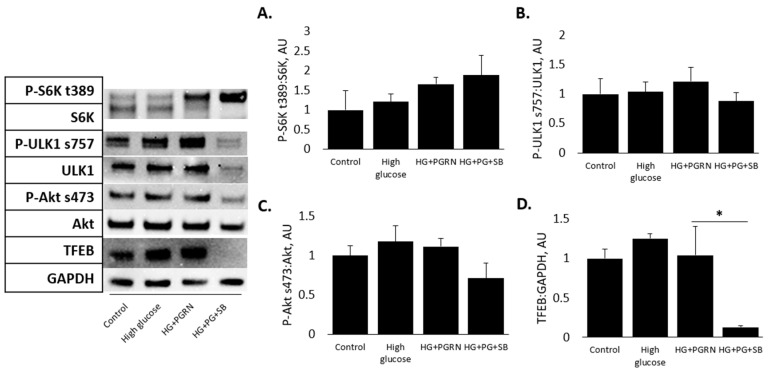
mTOR and Akt pathway phosphorylation do not appear to be affected by PGRN under high glucose in cortical cells. Cells were treated for 72 h with high glucose (HG, 100 mM glucose), 2.94 nM PGRN, and 10 µM SB216763 (SB) before fixing and imaging. Phosphorylation of S6K at thr 389 (**A**), ULK1 at serine 757 (**B**), and Akt at serine 473 (**C**) were unaffected by treatments. Total levels of TFEB (**D**) decreased after SB216763 treatment under HG + PGRN, but not in other conditions. N = 3 samples (N = 6 samples for p:total ULK1). *, *p* < 0.05.

**Figure 6 cells-12-01803-f006:**
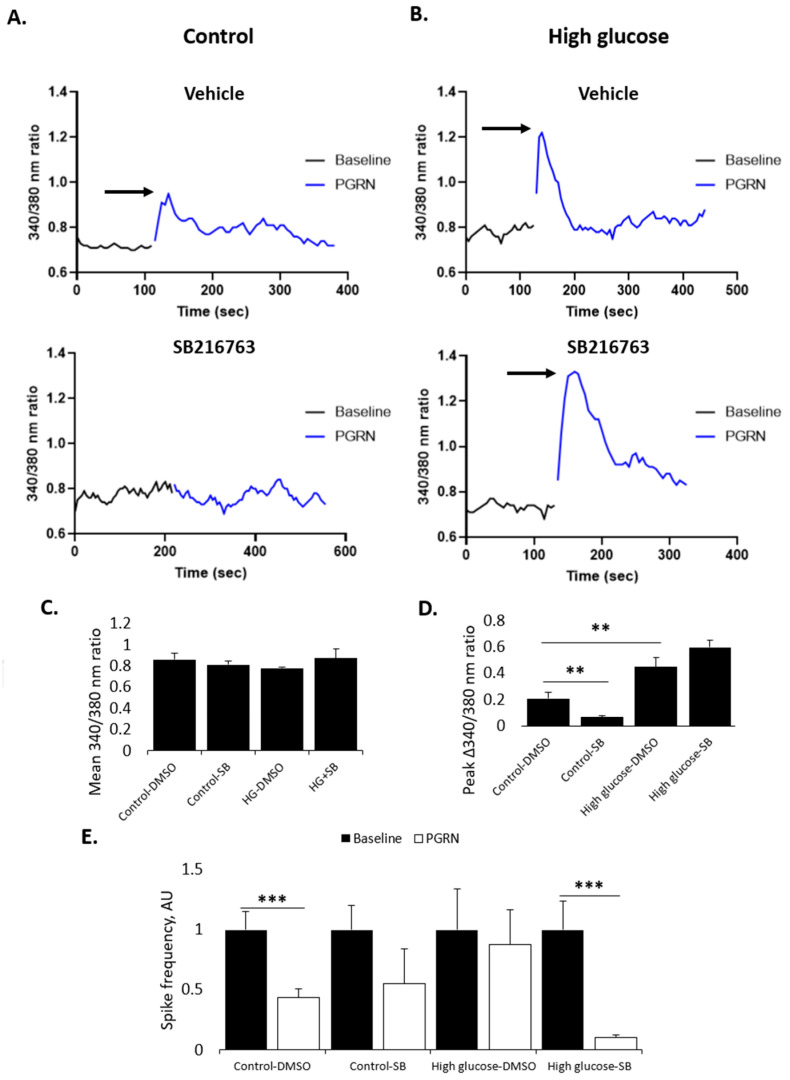
PGRN-mediated changes in intracellular calcium and neuronal firing are affected by GSK3β. Cells were treated for 72 h with normal or high glucose (HG, 100 mM glucose) medium before pre-treating with DMSO (vehicle) or 10 µM SB216763 (SB) 1 h prior to experimentation. Cells were incubated with Fura 2-AM calcium-binding ester for 30 min, then washed and imaged for fluorescence at 340 and 380 nm. Representative traces of neurons under normal (**A**) and high glucose (**B**) recorded before (Baseline) and after (PGRN) acute administration of 2.94 nM PGRN. Arrows indicate rapid spikes in calcium influx, as measured by a change in the 340/380 nm ratio. (**C**) The mean baseline 340/380 nm ratio was unchanged in all conditions. (**D**) The peak change in 340/380 nm ratio (Δ 340/380 nm ratio), seen after acute PGRN administration, was attenuated when neurons were pre-treated with SB216763, indicating reduced acute change in calcium influx in neurons. High glucose led to a greater Δ 340/380 nm ratio after acute PGRN administration that was not attenuated by SB216763 pre-treatment. N = 20–39 neurons from at least three experimental replicates. (**E**) MEA analysis showed decreased frequency of neuronal firing following acute PGRN administration that was attenuated by SB216763 pre-treatment. This effect was also observed in cells under high-glucose stress, which was reversed with SB216763 pre-treatment. N = over 100 channels from 3 experimental replicates. **, *p* < 0.01; ***, *p* < 0.001.

## Data Availability

The raw data supporting the conclusions of this paper will be made available by the authors on request with no undue reservation.

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
