# Peer review of "Progranulin Protects against Hyperglycemia-Induced Neuronal Dysfunction through GSK3β Signaling"

_cells, 2023, doi:10.3390/cells12131803_

Round 1

Reviewer 1 Report

The Dedert et al, have studied the mechanisms connecting Progranulin, hyperglycemia-induced neuronal dysfunction, and GSK3β signalling. The study is very interesting and well prepared.

This study presents compelling evidence indicating the involvement of GSK3β in facilitating the neuroprotective effects of PGRN (Progranulin) against high glucose-induced neuronal damage. The findings suggest that GSK3β contributes to the preservation of cell development, regulation of autophagy flux, and modulation of neuronal firing activity, thus highlighting its potential as a key player in mediating PGRN's protective mechanisms.

Abstract

The abstract was presented in a clear and comprehensive manner, effectively conveying the key concepts and findings.

Introduction and M&M

The introduction and M&M description was concise and effectively expressed the key points.

Results

I would like to clarify a few points at the results parts:

I would like to see an image of the TMRM staining in all treated cell cultures. Further the fluorescence images in Figure 3A are unclear. The LC3B staining is unclear, and cropping the images is not ideal. I prefer to view the entire image rather than a minimum of 40X magnification. The author can then crop the relevant portion to highlight the changes. If you have a better image available, it would be preferable for the reader. For such a sensitive staining of LC3B, the best approach is to use confocal microscopy, which provides better resolution. Please provide an overview of the single neuron without cutting it.

Discussion

Regarding the discussion part I have few comments and question as follows:

Did the author plan to investigate other signaling pathways?

Did the author observe any protein accumulation related to autophagy in their cell culture?"

I'm also curious if you checked for the presence of amyloid beta in your cell culture. Evidence suggests that GSK3β activation in T2D may promote the production and accumulation of Aβ. Moreover, Aβ itself can induce GSK3β activation, establishing a bidirectional relationship between GSK3β and Aβ. This interaction may contribute to the development of cognitive impairment and neurodegeneration observed in individuals with both T2D and AD.

What would be the effect of cellular starvation on these results?

What is your opinion, and what strategies do you suggest controlling this situation in neurodegenerative diseases and T2D?

Reviewer 2 Report

This is a nicely presented in vitro study using primary isolated cells rather than cell lines.  The experiments appear well done and the findings of interest.  The significance of progranulin to neurodegenerative diseases has not been fully appreciated but with its multifunctional roles as a neurotrophic factor as well as an anti-inflammatory, it has an interesting potential as a therapeutic agent.

This study covered the key properties of progranulin on neurons from several experimental angles.

Major Point

1.  It seems to me that these experiments need a protein control to show PGRN specificity.  Another protein of similar size and properties could be used, or using blocking antibodies to PGRN.  Even a carrier protein.  Comments?

Minor points.

1. I would suggest changing the figures and text when p>0.05 to indicate not significant.  There is little point of stating p<0.058 or  <0.080.  These values do not change the fact that the differences are not significant.

2.  Can you express doses of progranulin in nM as well.  This is useful to show physiological significance.

3.  There have been several recent publications on progranulin in human AD brains that were not referenced.  I think a more detailed discussion on the significance of the findings in relation to the disease would strengthen the readability of the paper and conclusions.  In addition, l am surprised there was no reference to Gao, 2010, and Wang 2022 whose findings are relevant to this publication

4.  Related to point 3, a lot of the discussion is not relevant and is speculative to the simple findings.  As the data is simple, a straight forward discussion of findings, and then fitting in relation to disease would make this a better paper.

5.  Please outline how doses and timepoints were determined.

Reviewer 3 Report

This manuscript addresses an interesting topic for the scientific community. However, the authors must proceed with a review of the article.

The way the authors address their findings about GSK3β  is not the best, as GSK3 can be inhibited by the phosphorylation of AKT. As diabetic individuals and diabetic animal models or in vitro models will also present insulin resistance or even nearly a total inhibition of the PI3K pathway this will lead to reduced phosphorylation of AKT, which will lead to higher phosphorylation of GSK3β . There are hundreds of publications about this in scientific literature. Therefore, the authors must find a better way to present their results other than saying that GSK3β  is a ¨candidate protein¨, and this expression is not even correct as proteins like Synaptophysin or PSD95 will present higher or lower expression levels, for example, whereas proteins that present sites of phosphorylation you must always refer as higher or lower phosphorylation, as the total expression levels of GSK3 or any other protein will not give you a significant information about your results.

There are articles showing that activation of mTOR2 by AMPK can lead to increased AKT phosphorylation, which would also inhibit GSK3β  phosphorylation. Therefore, the authors’ hypothesis in item 3.1 must be reviewed, and the authors will need to add information to this manuscript instead of speculation.

Figure 1B and 1C needs to be improved. There is no need for the chart below the graph. The figure needs to be done and presented in a better way. The same for the rest of the figures where the author uses a chart below the graph along the entire manuscript.

Y axis must be presented on all graphs.

The authors mention along the entire manuscript the importance of mTOR and AMPK, as it would be a major regulator for GSK3β , but they never presented any data about mTOR, and neither about AMPK. All your results go around speculation about mTOR1, since figure 1 (results), but the authors do not present any data about this. The authors should present extra data showing that it is the activation of mTOR1 , and not mTOR2, that is leading to the increased phosphorylation of GSK3β. There are many recent papers (original and reviews) that have been showing different roles for mTOR and AMPK than the “classic” role that the authors mention, but never show in their results, do not always happen, meaning the science has been changing about this. Presenting the results for total expression levels of mTOR, and its phosphorylation results in mTOR1 and mTOR2 is a must to complete the rationale of this paper. The authors should be able to get these results in 3 days through western blot. Presenting the AMPK results (optional) would also help the authors to explore better the mitochondrial results.

As the authors focus a lot of their writing in autophagy, but present a very small amount of data with LC3B and LAMP2A, the authors should add a western blot, with high quality, for TFEB, as this is a major regulator of autophagy pathway. ULK1 and its phosphorylation site must be done as it is activated by mTOR1, and sometimes from AMPK too. Please don’t do another immunofluorescence as they take longer time to be done and the images in the current manuscript, besides being “ok”, do not present the best quality. These results together will connect much better and in a much more rational way the entire speculation about mTOR and AMPK along the entire manuscript, and instead of results connected to speculation the authors will have solid results to present.

Please, review the entire paper and make the necessary modifications

Round 2

Reviewer 2 Report

Good revisions

Reviewer 3 Report

The authors have addressed the requests.